# Collecting Telemetry Data Privately

**Bolin Ding, Janardhan Kulkarni, Sergey Yekhanin**
Microsoft Research
{bolind, jakul, yekhanin}@microsoft.com

## Abstract

The collection and analysis of telemetry data from user's devices is routinely performed by many software companies. Telemetry collection leads to improved user experience but poses significant risks to users' privacy. Locally differentially private (LDP) algorithms have recently emerged as the main tool that allows data collectors to estimate various population statistics, while preserving privacy. The guarantees provided by such algorithms are typically very strong for a single round of telemetry collection, but degrade rapidly when telemetry is collected regularly. In particular, existing LDP algorithms are not suitable for repeated collection of counter data such as daily app usage statistics. In this paper, we develop new LDP mechanisms geared towards repeated collection of counter data, with formal privacy guarantees even after being executed for an arbitrarily long period of time. For two basic analytical tasks, mean estimation and histogram estimation, our LDP mechanisms for repeated data collection provide estimates with comparable or even the same accuracy as existing single-round LDP collection mechanisms. We conduct empirical evaluation on real-world counter datasets to verify our theoretical results. Our mechanisms have been deployed by Microsoft to collect telemetry across millions of devices.

## 1 Introduction

Collecting telemetry data to make more informed decisions is a commonplace. In order to meet users' privacy expectations and in view of tightening privacy regulations (e.g., European GDPR law) the ability to collect telemetry data privately is paramount. Counter data, e.g., daily app or system usage statistics reported in seconds, is a common form of telemetry. In this paper we are interested in algorithms that preserve users' privacy in the face of continuous collection of counter data, are accurate, and scale to populations of millions of users.

Recently, differential privacy [10] (DP) has emerged as defacto standard for the privacy guarantees. In the context of telemetry collection one typically considers algorithms that exhibit differential privacy in *the local model* [12, 14, 7, 5, 3, 18], also called randomized response model [19], $\gamma$-amplification [13], or FRAPP [1]. These are randomized algorithms that are invoked on each user's device to turn user's private value into a response that is communicated to a data collector and have the property that the likelihood of any specific algorithm's output varies little with the input, thus providing users with plausible deniability. Guarantees offered by locally differentially private algorithms, although very strong in a single round of telemetry collection, quickly degrade when data is collected over time. This is a very challenging problem that limits the applicability of DP in many contexts.

In telemetry applications, privacy guarantees need to hold in the face of continuous data collection. An influential paper [12] proposed a framework based on *memoization* to tackle this issue. Their techniques allow one to extend single round DP algorithms to continual data collection and protect users whose values stay constant or change very rarely. The key limitation of the work of [12] is that their approach cannot protect users' private numeric values with very small but frequent changes, making it inappropriate for collecting telemetry counters. In this paper, we address this limitation.

*We design mechanisms with formal privacy guarantees in the face of continuous collection of counter data. These guarantees are particularly strong when user's behavior remains approximately the same, varies slowly, or varies around a small number of values over the course of data collection.*

**Our results.** Our contributions are threefold.

1) We give simple 1-bit response mechanisms in the local model of DP for single-round collection of counter data for mean and histogram estimation. Our mechanisms are inspired by those in [19, 8, 7, 4], but allow for considerably simpler descriptions and implementations. Our experiments also demonstrate their performance in concrete settings.

2) Our main technical contribution is a rounding technique called $\alpha$-point rounding that borrows ideas from approximation algorithms literature [15, 2], and allows memoization to be applied in the context of private collection of counters. Our memoization schema avoids substantial losses in accuracy or privacy and unaffordable storage overhead. We give a rigorous definition of privacy guarantees provided by our algorithms when the data is collected continuously for an arbitrarily long period of time. We also present empirical findings related to our privacy guarantees.

3) Finally, our mechanisms have been deployed by Microsoft across millions of devices starting with Windows Insiders in Windows 10 Fall Creators Update to protect users' privacy while collecting application usage statistics.

## 1.1 Preliminaries and problem formulation

In our setup, there are $n$ *users*, and each user at time $t$ has a private (integer or real-valued) counter with value $x_i(t) \in [0, m]$. A *data collector* wants to collect these counter values $\{x_i(t)\}_{i \in [n]}$ at each time stamp $t$ to do statistical analysis. For example, for the telemetry analysis, understanding the mean and the distribution of counter values (e.g., app usage) is very important to IT companies.

**Local model of differential privacy (LDP).** Users do not need to trust the data collector and require formal privacy guarantees before they are willing to communicate their values to the data collector. Hence, a more well-studied DP model [10, 11], which first collects all users' data and then injects noise in the analysis step, is not applicable in our setup. In this work, we adopt the *local model of differential privacy*, where each user randomizes private data using a randomized algorithm (mechanism) $\mathcal{A}$ locally before sending it to data collector.

**Definition 1** ([13, 8, 4])**.** *A randomized algorithm $\mathcal{A} : \mathcal{V} \to \mathcal{Z}$ is $\epsilon$-locally differentially private ($\epsilon$-LDP) if for any pair of values $v, v' \in \mathcal{V}$ and any subset of output $S \subseteq \mathcal{Z}$, we have that*

$$\mathbf{Pr}[\mathcal{A}(v) \in S] \le e^\epsilon \cdot \mathbf{Pr}[\mathcal{A}(v') \in S].$$

LDP formalizes a type of plausible deniability: no matter what output is released, it is approximately equally as likely to have come from one point $v \in \mathcal{V}$ as any other. For alternate interpretations of differential privacy within the framework of hypothesis testing we refer the reader to [20, 7].

**Statistical estimation problems.** We focus on two estimation problems in this paper.

*Mean estimation:* For each time stamp $t$, the data collector wants to obtain an estimation $\hat{\sigma}(t)$ for the mean of $\vec{x_t} = \langle x_i(t) \rangle_{i \in [n]}$, i.e., $\sigma(\vec{x_t}) = \frac{1}{n} \cdot \sum_{i \in [n]} x_i(t)$. We do worst case analysis and aim to bound the *absolute error* $|\hat{\sigma}(t) - \sigma(\vec{x_t})|$ for any input $\vec{x_t} \in [0, m]^n$. In the rest of the paper, we abuse notation and denote $\sigma(t)$ to mean $\sigma(\vec{x_t})$ for a fixed input $\vec{x_t}$.

*Histogram estimation:* Suppose the domain of counter values is partitioned into $k$ buckets (e.g., with equal widths), and a counter value $x_i(t) \in [0, m]$ can be mapped to a bucket number $v_i(t) \in [k]$. For each time stamp $t$, the data collector wants to estimate frequency of $v \in [k] : h_t(v) = \frac{1}{n} \cdot |\{i : v_i(t) = v\}|$ as $\hat{h}_t(v)$. The *error* of a histogram estimation is measured by $\max_{v \in [k]} |\hat{h}_t(v) - h_t(v)|$. Again, we do worst case analysis of our algorithm over all possible inputs $\vec{v_t} = \langle v_i(t) \rangle_{i \in [n]} \in [k]^n$.

## 1.2 Repeated collection and overview of privacy framework

**Privacy leakage in repeated data collection.** Although LDP is a very strict notion of privacy, its effectiveness decreases if the data is collected repeatedly. If we collect counter values of a user $i$ for $T$ time stamps by executing an $\varepsilon$-LDP mechanism $\mathcal{A}$ independently on each time stamp,

$x_i(1)x_i(2)\ldots x_i(T)$ can be only guaranteed indistinguishable to another sequence of counter values, $x_i'(1)x_i'(2)\ldots x_i'(T)$, by a factor of up to $e^{T\cdot\varepsilon}$, which is too large to be reasonable as $T$ increases.

Hence, in applications such as telemetry, where data is collected continuously, privacy guarantees provided by an LDP mechanism for a single round of data collection are not sufficient. We formalize our privacy guarantee to enhance LDP for repeated data collection later in Section 3. However, intuitively we ensure that every user blends with a large set of other users who have very different behaviors.

**Our Privacy Framework and Guarantees.** Our framework for repeated private collection of counter data follows similar outline as the framework used in [12]. Our framework for mean and histogram estimation has four main components:

1) An important building block for our overall solution are 1-bit mechanisms that provide local $\epsilon$-LDP guarantees and good accuracy for a single round of data collection (Section 2).

2) An $\alpha$-point rounding scheme to randomly discretize users private values prior to applying memoization (to conceal small changes) while keeping the expectation of discretized values intact (Section 3).

3) Memoization of discretized values using the 1-bit mechanisms to avoid privacy leakage from repeated data collection (Section 3). In particular, if the counter value of a user remains approximately consistent, then the user is guaranteed $\epsilon$-differential privacy even after many rounds of data collection.

4) Finally, output perturbation (instantaneous noise in [12]) to protect exposing the transition points due to large changes in user's behavior and attacks based on auxiliary information (Section 4).

In Sections 2, 3 and 4, we formalize these guarantees focusing predominantly on the mean estimation problem. All the omitted proofs and additional experimental results are in the full version on the arXiv [6].

## 2 Single-round LDP mechanisms for mean and histogram estimation

We first describe our 1-bit LDP mechanisms for mean and histogram estimation. Our mechanisms are inspired by the works of Duchi *et al.* [8, 7, 9] and Bassily and Smith [4]. However, our mechanisms are tuned for more efficient communication (by sending 1 bit for each counter each time) and stronger protection in repeated data collection (introduced later in Section 3). To the best our knowledge, the exact form of mechanisms presented in this Section was not known. Our algorithms yield accuracy gains in concrete settings (see Section 5) and are easy to understand and implement.

### 2.1 1-Bit mechanism for mean estimation

**Collection mechanism** 1BitMean: When the collection of counter $x_i(t)$ at time $t$ is requested by the data collector, each user $i$ sends one bit $b_i(t)$, which is independently drawn from the distribution:

$$b_i(t) = \begin{cases} 1, & \text{with probability } \frac{1}{e^\epsilon+1} + \frac{x_i(t)}{m} \cdot \frac{e^\epsilon-1}{e^\epsilon+1}; \\ 0, & \text{otherwise.} \end{cases} \tag{1}$$

**Mean estimation.** Data collector obtains the bits $\{b_i(t)\}_{i\in[n]}$ from $n$ users and estimates $\sigma(t)$ as

$$\hat{\sigma}(t) = \frac{m}{n}\sum_{i=1}^{n}\frac{b_i(t)\cdot(e^\varepsilon+1)-1}{e^\varepsilon-1}. \tag{2}$$

The basic randomizer of [4] is equivalent to our 1-bit mechanism for the case when each user takes values either $0$ or $m$. The above mechanism can also be seen as a simplification of the multidimensional mean-estimation mechanism given in [7]. For the 1-dimensional mean estimation, Duchi *et al.* [7] show that Laplace mechanism is *asymptotically optimal* for the mini-max error. However, the communication cost per user in Laplace mechanism is $\Omega(\log m)$ bits, and our experiments show it also leads to larger error compared to our 1-bit mechanism. We prove following results for the above 1-bit mechanism.

**Theorem 1.** *For single-round data collection, the mechanism* 1BitMean *in* (1) *preserves $\epsilon$-LDP for each user. Upon receiving the $n$ bits $\{b_i(t)\}_{i\in[n]}$, the data collector can then estimate the mean of*

*counters from $n$ users as $\hat{\sigma}(t)$ in (2). With probability at least $1 - \delta$, we have*

$$|\hat{\sigma}(t) - \sigma(t)| \le \frac{m}{\sqrt{2n}} \cdot \frac{e^\varepsilon + 1}{e^\varepsilon - 1} \cdot \sqrt{\log \frac{2}{\delta}}.$$

## 2.2 $d$-Bit mechanism for histogram estimation

Now we consider the problem of estimating histograms of counter values in a discretized domain with $k$ buckets with LDP to be guaranteed. This problem has extensive literature both in computer science and statistics, and dates back to the seminal work Warner [19]; we refer the readers to following excellent papers [16, 8, 4, 17] for more information. Recently, Bassily and Smith [4] gave asymptotically tight results for the problem in the worst-case model building on the works of [16]. On the other hand, Duchi *et al.* [8] introduce a mechanism by adapting Warner's classical randomized response mechanism in [19], which is shown to be optimal for the statistical mini-max regret if one does not care about the cost of communication. Unfortunately, some ideas in Bassily and Smith [4] such as Johnson-Lindenstrauss lemma do not scale to population sizes of millions of users. Therefore, in order to have a smooth trade-off between accuracy and communication cost (as well as the ability to protect privacy in repeated data collection, which will be introduced in Section 3) we introduce a modified version of Duchi *et al.*'s mechanism [8] based on subsampling by buckets.

**Collection mechanism $d$BitFlip:** Each user $i$ randomly draws $d$ bucket numbers without replacement from $[k]$, denoted by $j_1, j_2, \ldots, j_d$. When the collection of discretized bucket number $v_i(t) \in [k]$ at time $t$ is requested by the data collector, each user $i$ sends a vector:

$$b_i(t) = [(j_1, b_{i,j_1}(t)), (j_2, b_{i,j_2}(t)), \ldots, (j_d, b_{i,j_d}(t))], \text{ where } b_{i,j_p}(t) \text{ is a random 0-1 bit,}$$

$$\text{with } \mathbf{Pr}\big[b_{i,j_p}(t) = 1\big] = \begin{cases} e^{\varepsilon/2}/(e^{\varepsilon/2} + 1) & \text{if } v_i(t) = j_p \\ 1/(e^{\varepsilon/2} + 1) & \text{if } v_i(t) \ne j_p \end{cases}, \text{ for } p = 1, 2, \ldots, d.$$

Under the same public coin model as in [4], each user $i$ only needs to send to the data collector $d$ bits $b_{i,j_1}(t), b_{i,j_2}(t), \ldots, b_{i,j_d}(t)$ in $b_i(t)$, as $j_1, j_2, \ldots, j_d$ can be generated using public coins.

**Histogram estimation.** Data collector estimates histogram $h_t$ as: for $v \in [k]$,

$$\hat{h}_t(v) = \frac{k}{nd} \sum_{b_{i,v}(t) \text{ is received}} \frac{b_{i,v}(t) \cdot (e^{\varepsilon/2} + 1) - 1}{e^{\varepsilon/2} - 1}. \tag{3}$$

When $d = k$, $d$BitFlip is exactly the same as the one in Duchi *et al.*[8]. The privacy guarantee is straightforward. In terms of the accuracy, the intuition is that for each bucket $v \in [k]$, there are roughly $nd/k$ users responding with a 0-1 bit $b_{i,v}(t)$. We can prove the following result.

**Theorem 2.** *For single-round data collection, the mechanism $d$BitFlip preserves $\epsilon$-LDP for each user. Upon receiving the $d$ bits $\{b_{i,j_p}(t)\}_{p \in [d]}$ from each user $i$, the data collector can then estimate then histogram $h_t$ as $\hat{h}_t$ in (3). With probability at least $1 - \delta$, we have,*

$$\max_{v \in [k]} |h_t(v) - \hat{h}_t(v)| \le \sqrt{\frac{5k}{nd}} \cdot \frac{e^{\varepsilon/2} + 1}{e^{\varepsilon/2} - 1} \cdot \sqrt{\log \frac{6k}{\delta}} \le \mathrm{O}\left(\sqrt{\frac{k \log(k/\delta)}{\varepsilon^2 nd}}\right).$$

# 3 Memoization for continual collection of counter data

One important concern regarding the use of $\epsilon$-LDP algorithms (e.g., in Section 2.1) to collect counter data pertains to privacy leakage that may occur if we collect user's data repeatedly (say, daily) and user's private value $x_i$ does not change or changes little. Depending on the value of $\epsilon$, after a number of rounds, data collector will have enough noisy reads to estimate $x_i$ with high accuracy.

Memoization [12] is a simple rule that says that: *At the account setup phase each user pre-computes and stores his responses to data collector for all possible values of the private counter. At data collection users do not use fresh randomness, but respond with pre-computed responses corresponding to their current counter values.* Memoization (to a certain degree) takes care of situations when the private value $x_i$ stays constant. Note that the use of memoization violates differential privacy in continual collection. If memoization is employed, data collector can easily distinguish a user whose

value keeps changing, from a user whose value is constant; no matter how small the $\epsilon$ is. However, privacy leakage is limited. When data collector observes that user's response had changed, this only indicates that user's value had changed, but not what it was and not what it is.

As observed in [12, Section 1.3] using memoization technique in the context of collecting counter data is problematic for the following reason. Often, from day to day, private values $x_i$ do not stay constant, but rather experience small changes (e.g., one can think of app usage statistics reported in seconds). Note that, naively using memoization adds no additional protection to the user whose private value varies but stays approximately the same, as data collector would observe many independent responses corresponding to it.

One naive way to fix the issue above is to use discretization: pick a large integer (segment size) $s$ that divides $m$; consider the partition of all integers into segments $[\ell s, (\ell+1)s]$; and have each user report his value after rounding the true value $x_i$ to the mid-point of the segment that $x_i$ belongs to. This approach takes care of the issue of leakage caused by small changes to $x_i$ as users values would now tend to stay within a single segment, and thus trigger the same memoized response; however accuracy loss may be extremely large. For instance, in a population where all $x_i$ are $\ell s + 1$ for some $\ell$, after rounding every user would be responding based on the value $\ell s + s/2$.

In the following subsection we present a better (randomized) rounding technique (termed $\alpha$-point rounding) that has been previously used in approximation algorithms literature [15, 2] and rigorously addresses the issues discussed above. We first consider the mean estimation problem.

### 3.1 $\alpha$-point rounding for mean estimation

The key idea of rounding is to discretize the domain where users' counters take their values. Discretization reduces domain size, and users that behave consistently take less different values, which allows us to apply memoization to get a strong privacy guarantee.

As we demonstrated above discretization may be particularly detrimental to accuracy when users' private values are correlated. We propose addressing this issue by: *making the discretization rule independent across different users*. This ensures that when (say) all users have the same value, some users round it up and some round it down, facilitating a smaller accuracy loss.

We are now ready to specify the algorithm that extends the basic algorithm 1BitMean and employs both $\alpha$-point rounding and memoization. We assume that counter values range in $[0, m]$.

1. At the algorithm design phase, we specify an integer $s$ (our discretization granularity). We assume that $s$ divides $m$. We suggest setting $s$ rather large compared to $m$, say $s = m/20$ or even $s = m$ depending on the particular application domain.

2. At the the setup phase, each user $i \in [n]$ independently at random picks a value $\alpha_i \in \{0, \dots, s-1\}$, that is used to specify the rounding rule.

3. User $i$ invokes the basic algorithm 1BitMean with range $m$ to compute and memoize 1-bit responses to data collector for all $\frac{m}{s} + 1$ values $x_i$ in the arithmetic progression

$$A = \{\ell s\}_{0 \le \ell \le \frac{m}{s}}. \tag{4}$$

4. Consider a user $i$ with private value $x_i$ who receives a data collection request. Let $x_i \in [L, R)$, where $L, R$ are the two neighboring elements of the arithmetic progression $\{\ell s\}_{0 \le \ell \le \frac{m}{s}+1}$. The user $x_i$ rounds value to $L$ if $x_i + \alpha_i < R$; otherwise, the user rounds the value to $R$. Let $y_i$ denote the value of the user after rounding. In each round, user responds with the memoized bit for value $y_i$. Note that rounding is always uniquely defined.

Perhaps a bit surprisingly, using $\alpha$-point rounding does not lead to additional accuracy losses independent of the choice of discretization granularity $s$.

**Theorem 3.** *Independent of the value of discretization granularity $s$, at any round of data collection, each output bit $b_i$ is still sampled according to the distribution given by formula (1). Therefore, the algorithm above provides the same accuracy guarantees as given in Theorem 1.*

## 3.2 Privacy definition using permanent memoization

In what follows we detail privacy guarantees provided by an algorithm that employs $\alpha$-point rounding and memoization in conjunction with the $\epsilon$-DP 1-bit mechanism of Section 2.1 against a data collector that receives a very long stream of user's responses to data collection events.

Let $U$ be a user and $x(1), \ldots, x(T)$ be the sequence of $U$'s private counter values. Given user's private value $\alpha_i$, each of $\{x(j)\}_{j \in [T]}$ gets rounded to the corresponding value $\{y(j)\}_{j \in [T]}$ in the set $A$ (defined by (4)) according to the rule given in Section 3.1.

**Definition 2.** *Let $B$ be the space of all sequences $\{z(j)\}_{j \in [T]} \in A^T$, considered up to an arbitrary permutation of the elements of $A$. We define the behavior pattern $b(U)$ of the user $U$ to be the element of $B$ corresponding to $\{y(j)\}_{j \in [T]}$. We refer to the number of distinct elements $y(j)$ in the sequence $\{y(j)\}_{j \in [T]}$ as the width of $b(U)$.*

We now discuss our notion of behavior pattern, using counters that carry daily app usage statistics as an example. Intuitively, users map to the same behavior pattern if they have the same number of different modes (approximate counter values) of using the app, and switch between these modes on the same days. For instance, one user that uses an app for 30 minutes on weekdays, 2 hours on weekends, and 6 hours on holidays, and the other user who uses the app for 4 hours on weekdays, 10 minutes on weekends, and does not use it on holidays will likely map to the same behavior pattern. Observe however that the mapping from actual private counter values $\{x(j)\}$ to behavior patterns is randomized, thus there is a likelihood that some users with identical private usage profiles may map to different behavior patterns. This is a positive feature of the Definition 2 that increases entropy among users with the same behavior pattern.

The next theorem shows that the algorithm of Section 3.1 makes users with the same behavior pattern blend with each other from the viewpoint of data collector (in the sense of differential privacy).

**Theorem 4.** *Consider users $U$ and $V$ with sequences of private counter values $\{x_U(1), \ldots, x_U(T)\}$ and $\{x_V(1), \ldots, x_V(T)\}$. Assume that both $U$ and $V$ respond at $T$ data-collection time stamps using the algorithm presented in Section 3.1, and $b(U) = b(V)$ with the width of $b(U)$ equal to $w$. Let $\mathbf{s}_U, \mathbf{s}_V \in \{0,1\}^T$ be the random sequences of responses generated by users $U$ and $V$; then for any binary string $\mathbf{s} \in \{0,1\}^T$ in the response domain, we have:*

$$\Pr[\mathbf{s}_U = \mathbf{s}] \leq e^{w\epsilon} \cdot \Pr[\mathbf{s}_V = \mathbf{s}].  \tag{5}$$

### 3.2.1 Setting parameters

The $\epsilon$-LDP guarantee provided by Theorem 4 ensures that each user is indistinguishable from other users with the same behavior pattern (in the sense of LDP). The exact shape of behavior patterns is governed by the choice of the parameter $s$. Setting $s$ very large, say $s = m$ or $s = m/2$ reduces the number of possible behavior patterns and thus increases the number of users that blend by mapping to a particular behavior pattern. It also yields stronger guarantee for blending within a pattern since for all users $U$ we necessarily have $b(U) \leq m/s + 1$ and thus by Theorem 4 the likelihood of distinguishing users within a pattern is trivially at most $e^{(m/s+1) \cdot \epsilon}$. At the same time there are cases where one can justify using smaller values of $s$. In fact, consistent users, i.e., users whose private counter always land in the vicinity of one of a small number of fixed values enjoy a strong LDP guarantee within their patterns irrespective of $s$ (provided it is not too small), and smaller $s$ may be advantageous to avoid certain attacks based on auxiliary information as the set of all possible values of a private counter $x_i$ that lead to a specific output bit $b$ is potentially more complex.

Finally, it is important to stress that the $\epsilon$-LDP guarantee established in Theorem 4 is not a panacea, and in particular it is a weaker guarantee provided in a much more challenging setting than just the $\epsilon$-LDP guarantee across all users that we provide for a single round of data collection (an easier setting). In particular, while LDP across all population of users is resilient to any attack based on auxiliary information, LDP across a sub population may be vulnerable to such attacks and additional levels of protection may need to be applied. In particular, if data collector observes that user's response has changed; data collector knows with certainty that user's true counter value had changed. In the case of app usage telemetry this implies that app has been used on one of the days. This attack is partly mitigated by the output perturbation technique that is discussed in Section 4.

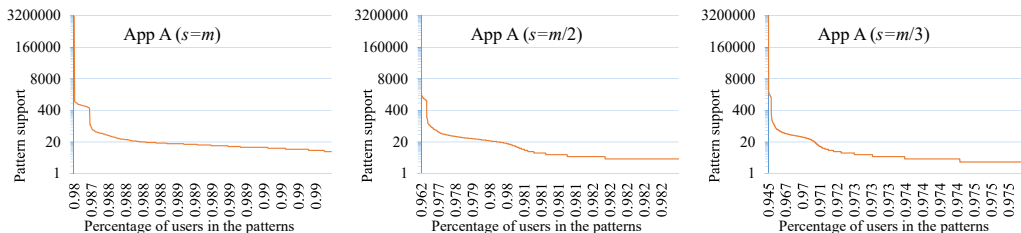
Figure 1: Distribution of pattern supports for App A

### 3.2.2 Experimental study

We use a real-world dataset of 3 million users with their daily usage of an app (App A) collected (in seconds) over a continuous period of 31 days to demonstrate the mapping of users to behavior patterns in Figure 1. See full version of the paper for usage patterns for more apps. For each behavior pattern (Definition 2), we calculate its *support* as *the number of users with their sequences in this pattern*. All the patterns' supports $sup$ are plotted ($y$-axis) in the decreasing order, and we can also calculate the percentage of users ($x$-axis) in patterns with supports at least $sup$. We vary the parameter $s$ in permanent memoization from $m$ (maximizing blending) to $m/3$ and report the corresponding distributions of pattern supports in Figure 1.

It is not hard to see that theoretically for every behavior pattern there is a very large set of sequences of private counter values $\{x(t)\}_t$ that may map to it (depending on $\alpha_i$). Real data (Figure 1) provides evidence that users tend to be approximately consistent and therefore simpler patterns, i.e., patterns that mostly stick to a single rounded value $y(t) = y$ correspond to larger sets of sequences $\{x_i(t)\}_t$, obtained from a real population. In particular, for each app there is always one pattern (corresponding to having one fixed $y(t) = y$ across all 31 days) which blends the majority of users ($> 2$ million). More complex behavior patterns have less users mapping to them. In particular, there always are some lonely users (1%-5% depending on $s$) who land in patterns that have support size of one or two. From the viewpoint of a data collector such users can only be identified as those having a complex and irregular behavior, however the actual nature of that behavior by Theorem 4 remains uncertain.

### 3.3 Example

One specific example of a counter collection problem that has been identified in [12, Section 1.3] as being non-suitable for techniques presented in [12] but can be easily solved using our methods is to repeatedly collect age in days from a population of users. When we set $s = m$ and apply the algorithm of Section 3.1 we can collect such data for $T$ rounds with high accuracy. Each user necessarily responds with a sequence of bits that has form $z^t \circ \bar{z}^{T-t}$, where $0 \leq t \leq T$. Thus data collector only gets to learn the transition point, i.e., the day when user's age in days passes the value $m - \alpha_i$, which is safe from privacy perspective as $\alpha_i$ is picked uniformly at random by the user.

### 3.4 Continual collection for histogram estimation using permanent memoization

**Naive memoization.** $\alpha$-point rounding is not suitable for histogram estimation as counter values have been mapped to $k$ buckets. The single-round LDP mechanism in Duchi *et al.* [8] sends a 0-1 random response for each bucket: send 1 with probability $e^{\varepsilon/2}/(e^{\varepsilon/2}+1)$ if the value is in this bucket, and with probability $1/(e^{\varepsilon/2}+1)$ if not. This mechanism is $\epsilon$-LDP. Each user can then memoize a mapping $f_k : [k] \to \{0,1\}^k$ by running this mechanism once for each $v \in [k]$, and always respond $f_k(v)$ if the user's value is in bucket $v$. However, this memoization schema leads to serious privacy leakage: with some auxiliary information, one can infer with high confidence a user's value from the response produced by the mechanism; more concretely, if the data collector knows that the app usage value is in a bucket $v$ and observes the output $f_k(v) = z$ in some day, whenever the user sends $z$ again in future, the data collector can infer that the bucket number is $v$ with almost 100% probability.

$d$**-bit memoization.** To avoid such privacy leakages, we memoize based on our $d$-bit mechanism $d$BitFlip (Section 2.2). Each user runs $d$BitFlip for each $v \in [k]$, with responses created on $d$ buckets $j_1, j_2, \ldots, j_d$ (randomly drawn and then fixed per user), and memoizes the response in a mapping $f_d : [k] \to \{0,1\}^d$. A user will always send $f_d(v)$ if the bucket number is $v$. This mechanism is denoted by $d$BitFlipPM, and the same estimator (3) can be used to estimate the histogram upon

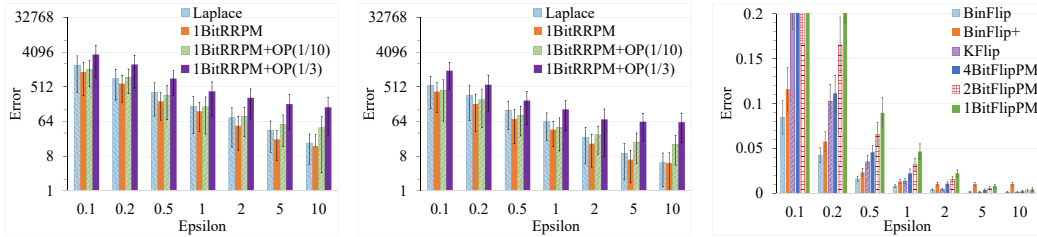

(a) Mean ($n = 0.3 \times 10^6$)  (b) Mean ($n = 3 \times 10^6$)  (c) Histogram ($n = 0.3 \times 10^6$)

Figure 2: Comparison of mechanisms for mean and histogram estimations on real-world datasets

receiving the $d$-bit response from every user. This scheme avoids privacy leakages that arise due to the naive memoization, because multiple ($\Omega(k/2^d)$ w.h.p.) buckets are mapped to the same response. This protection is the strongest when $d = 1$. Definition 2 about behavior patterns and Theorem 4 can be generalized here to provide similar privacy guarantee in continual data collection.

## 4 Output perturbation

One of the limitations of our memoization approach based on $\alpha$-point rounding is that it does not protect the points of time where user's behavior changes significantly. Consider a user who never uses an app for a long time, and then starts using it. When this happens, suppose the output produced by our algorithm changes from $0$ to $1$. Then the data collector can learn with certainty that the user's behavior changed, (but not what this behavior was or what it became). Output perturbation is one possible mechanism of protecting the exact location of the points of time where user's behavior has changed. As mentioned earlier, output perturbation was introduced in [12] as a way to mitigate privacy leakage that arises due to memoization. The main idea behind output perturbation is to flip the output of memoized responses with a small probability $0 \leq \gamma \leq 0.5$. This ensures that data collector will not be able to learn with certainty that the behavior of a user changes at certain time stamps. In the full version of the paper we formalize this notion, and prove accuracy and privacy guarantees with output perturbation. Here we contain ourselves to mentioning that using output perturbation with a positive $\gamma$, in combination with the $\epsilon$-LDP 1BitMean algorithm in Section 2 is equivalent to invoking the 1BitMean algorithm with $\epsilon' = \ln\left(\frac{(1-2\gamma)(\frac{e^\epsilon}{e^\epsilon+1})+\gamma}{(1-2\gamma)(\frac{1}{e^\epsilon+1})+\gamma}\right)$.

## 5 Empirical evaluation

We compare our mechanisms (with permanent memoization) for mean and histogram estimation with previous mechanisms for one-time data collection. We would like to emphasize that the goal of these experiments is to show that our mechanisms, with such additional protection, are no worse than or comparable to the state-of-the-art LDP mechanisms in terms of estimation accuracy.

We first use the real-world dataset which is described in Section 3.2.2.

**Mean estimation.** We implement our 1-bit mechanism (Section 2.1) with $\alpha$-point Randomized Rounding and Permanent Memoization for repeated collection (Section 3), denoted by 1BitR-RPM, and output perturbation to enhance the protection for usage change (Section 4), denoted by 1BitRRPM+OP($\gamma$). We compare it with the Laplace mechanism for LDP mean estimation in [8, 9], denoted by Laplace. We vary the value of $\varepsilon$ ($\varepsilon = 0.1\text{-}10$) and the number of users ($n = 0.3, 3 \times 10^6$ by randomly picking subsets of all users), and run all the mechanisms 3000 times on 31-day usage data with three counters. The domain size is $m = 24$ hours. The average of absolute errors (in seconds) with one standard deviation (STD) are reported in Figures 2(a)-2(b). 1BitRRPM is consistently better than Laplace with smaller errors and narrower STDs. Even with a perturbation probability $\gamma = 1/10$, they are comparable in accuracy. When $\gamma = 1/3$, output perturbation is equivalent to adding an additional uniform noise from $[0, 24 \text{ hours}]$ independently on each day–even in this case, 1BitRRPM+OP(1/3) gives us tolerable accuracy when the number of users is large.

**Histogram estimation.** We create $k = 32$ buckets on $[0, 24 \text{ hours}]$ with even widths to evaluate mechanisms for histogram estimation. We implement our $d$-bit mechanism (Section 2.2) with

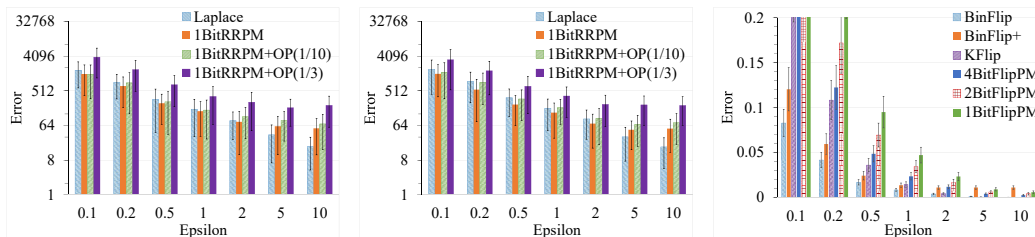

(a) Mean (constant distribution)  (b) Mean (uniform distribution)  (c) Histogram (normal distribution)

Figure 3: Mechanisms for mean and histogram estimations on different distributions ($n = 0.3 \times 10^6$)

permanent memoization for repeated collection (Section 3.4), denoted by $d$BitFlipPM. In order to provide protection on usage change in repeated collection, we use $d = 1, 2, 4$ (strongest when $d = 1$). We compare it with state-of-the-art one-time mechanisms for histogram estimation: BinFlip [8, 9], KFlip (k-RR in [17]), and BinFlip+ (applying the generic protocol with 1-bit reports in [4] on BinFlip). When $d = k$, $d$BitFlipPM has the same accuracy as BinFlip. KFlip is sub-optimal for small $\varepsilon$ [17] but has better performance when $\varepsilon$ is $\Omega(\ln k)$. In contrast, BinFlip+ has good performance when $\varepsilon \leq 2$. We repeat the experiment 3000 times and report the average *histogram error* (i.e., maximum error across all bars in a histogram) with one standard deviation for different algorithms in Figure 2(c) with $\varepsilon = 0.1\text{-}10$ and $n = 0.3 \times 10^6$ to confirm the above theoretical results. BinFlip (equivalently, 32BitFlipPM) has the best accuracy overall. With enhanced privacy protection in repeated data collection, 4bitFlipPM is comparable to the one-time collection mechanism KFlip when $\varepsilon$ is small (0.1-0.5) and 4bitFlipPM-1bitFlipPM are better than BinFlip+ when $\varepsilon$ is large (5-10).

**On different data distributions.** We have shown that errors in mean and histogram estimations can be bounded (Theorems 1-2) in terms of $\varepsilon$ and the number of users $n$, together with the number of buckets $k$ and the number of bits $d$ (applicable only to histograms). We now conduct additional experiments on synthetic datasets to verify that the empirical errors should not change much on different data distributions. Three types of distributions are considered: i) constant distribution, i.e., each user $i$ has a counter $x_i(t) = 12$ (hours) all the time; ii) uniform distribution, i.e., $x_i(t) \sim \mathcal{U}(0, 24)$; and iii) normal distribution, i.e., $x_i(t) \sim \mathcal{N}(12, 2^2)$ (with mean equal to 12 and standard deviation equal to 2), truncated on $[0, 24]$. Three synthetic datasets are created by drawing samples of sizes $n = 0.3 \times 10^6$ from these three distributions. Some results are plotted on Figure 3: the empirical errors on different distributions are almost the same as those in Figures 2(a) and 2(c). One can refer to the full version of the paper [6] for the complete set of charts.

## 6  Deployment

In earlier sections, we presented new LDP mechanisms geared towards repeated collection of counter data, with formal privacy guarantees even after being executed for a long period of time. Our mean estimation algorithm has been deployed by Microsoft starting with Windows Insiders in Windows 10 Fall Creators Update. The algorithm is used to collect the number of seconds that a user has spend using a particular app. Data collection is performed every 6 hours, with $\epsilon = 1$. Memoization is applied across days and output perturbation uses $\gamma = 0.2$. According to Section 4, this makes a single round of data collection satisfy $\epsilon'$-DP with $\epsilon' = 0.686$.

One important feature of our deployment is that collecting usage data for multiple apps from a single user only leads to a minor additional privacy loss that is independent of the actual number of apps. Intuitively, this happens since we are collecting active usage data, and the total number of seconds that a user can spend across multiple apps in 6 hours is bounded by an absolute constant that is independent of the number of apps.

**Theorem 5.** *Using the* 1BitMean *mechanism with a parameter $\epsilon'$ to simultaneously collect $t$ counters $x_1, \ldots, x_t$, where each $x_i$ satisfies $0 \leq x_i \leq m$ and $\sum_i x_i \leq m$ preserves $\epsilon''$-DP, where*

$$\epsilon'' = \epsilon' + e^{\epsilon'} - 1.$$

We defer the proof to the full version of the paper [6]. By Theorem 5, in deployment, a single round of data collection across an arbitrary large number of apps satisfies $\epsilon''$-DP, where $\epsilon'' = 1.672$.

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
