[Reviews · NeurIPS 2017]

Reviewer 1



Line 122 The authors claim that the 1-bit mechanism for mean estimation is tuned for efficient communication. Is there any difference between the basic randomizer in [3] with one item and the 1-bit mechanism? If the protocol is equivalent to the basic randomizer in [3] in a specific setting, it should be clearly stated. Theorem 2 Please compare the utility in terms of n and d with existing histogram protocols. Sec 3.1 It was unclear to me what privacy is guaranteed after alpha-point rounding. If a data collector reports her data repeatedly (say N times), the privacy budget should be set following the composition theorem. The alpha-point rounding introduces another random source into the responses and it certainly improves privacy to some extent, but the composition theorem still is applied as long as she responds to the data collector multiple times.

Reviewer 2



The authors propose a more efficient mechanism for local differential privacy. In particular, they propose a 1 bit local differential privacy method that enables mean estimation in a distributed and effective way. Also a d-bit method is proposed for histogram estimation. It is argued that these two goals are of partical relevance in collection of telemetry data. The main advance is in the communication efficiency - which is key to large scale deployment. The authors also note that this mechanism is indeed deployed at large scale in industry and likely consistutes the largest deployment of differential privacy to date. While this on its own does not add to the value of the scientific contribution - it does reinforce the point of an efficient, effective and practical approach. The authors submitted another version of their paper as supplementary material that contains the proofs. But it also contains more figures and extended main text. This is borderline. The supplementary material should only contain the additional material - while the main paper should be self-contained and not repeated. It is also weird, that the reviewer has to hunt for proofs in provided text. There should be a clear separation between submission and supplementary material. Overall, the presentation is clear - partially due to outsourcing parts of the submission to the supplementary. However, alpha-rounding part remains partially unclear. Can the authors please elaborate on the claimed guarantees? While the experimental study is interesting - there is a question about reproducibility of the results.As it looks this is real user and company data - how would follow up work compare here? It would be beneficial to provide an alternative solution -- potential simulated data -- so that comparison remain open source. However, the experiments are considered convincing and support the main claim.